# OOD-Chameleon: Is Algorithm Selection for OOD Generalization Learnable?

## Abstract

Out-of-distribution (OOD) generalization is challenging because distribution shifts come in many forms. A multitude of learning algorithms exist and each can improve performance in *specific* OOD situations. We posit that much of the challenge of OOD generalization lies in *choosing the right algorithm for the right dataset*. However, such algorithm selection is often elusive under complex real-world shifts. In this work, we formalize the task of *algorithm selection for OOD generalization* and investigate whether it could be approached by learning.

We propose a solution, dubbed OOD-Chameleon that formulates the task as a supervised classification over candidate algorithms. We construct a *dataset of datasets* to learn from, which represents diverse types, magnitudes and combinations of shifts (covariate shift, label shift, spurious correlations). We train the model to predict the relative performance of algorithms given a dataset's characteristics. This enables *a priori* selection of the best learning strategy, i.e. without training various models as needed with traditional model selection.

Our experiments show that the adaptive selection outperforms any individual algorithm and simple selection heuristics, on unseen datasets of controllable and realistic image data. Inspecting the model shows that it learns non-trivial data/algorithms interactions, and reveals the conditions for any one algorithm to surpass another. This opens new avenues for (1) enhancing OOD generalization with existing algorithms ~~instead of designing new ones~~, and (2) gaining insights into the applicability of existing algorithms with respect to datasets' properties.

## 1 Introduction

**The many faces of OOD generalization.** Out-of-distribution (OOD) generalization refers to a model's ability to remain accurate when the distributions of the training and test data differ. "OOD" is a catch-all term since it encompasses many types of distribution shifts (Wiles et al., 2021; Ye et al., 2022; Nagarajan et al., 2021). In medical imaging for example (Oakden-Rayner et al., 2020), a model may have to process X-rays from various demographics (*covariate shift*), pathologies (*label shift*), and co-occurrences of patient attributes (*spurious correlations*). These types of shifts have often been studied independently, and are often best addressed with different algorithms. However, real data is often complicated by *combinations* of shifts of different types and magnitudes (Wiles et al., 2021; Yang et al., 2023) that interact in complex ways with the learning algorithms (Jiang et al., 2023; Cabannes et al., 2023; Benoit et al., 2024).

**The trade-offs of learning algorithms.** OOD generalization is challenging because it is fundamentally underspecified (D'Amour et al., 2022; Teney et al., 2022). The training set alone is not enough to inform about the nature of the shifts nor to constrain the behavior of the model on OOD test data. This is why a multitude of learning algorithms exist, which rely on different assumptions or side information (e.g. domain labels), from standard ERM (Vapnik, 2000) to simple interventions such as Resampling, GroupDRO (Sagawa et al., 2020a), and more-complex ones (Liu et al., 2023).

A well-known study by Gulrajani & Lopez-Paz (2020) showed that none of the methods, at the time of their study, surpasses an ERM baseline across a collection of datasets. This is *not* surprising though: each dataset exhibits different characteristics and types or magnitudes of shifts that call for different methods (Benoit et al., 2024). Numerous newer analyses indeed confirm that (at the time of

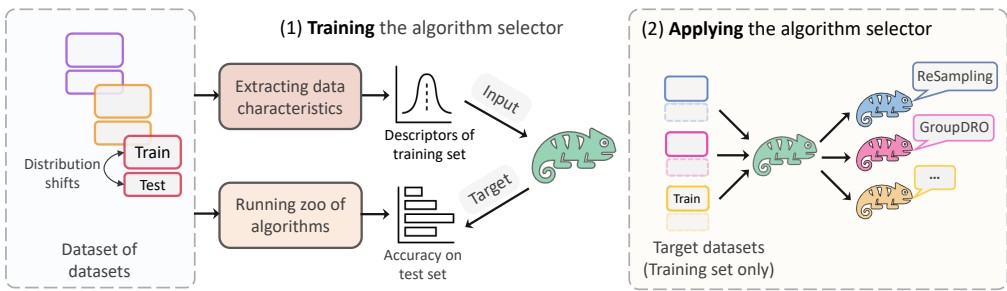

Figure 1: Given a dataset, how can one choose an algorithm such that the most robust model is obtained after training on the dataset? (**right**) We propose the task of **algorithm selection for OOD generalization** and build an automated algorithm selector dubbed OOD-CHAMELEON. We train the model on a dataset of datasets that exemplify a variety of distribution shifts (**left**) and show that it learns non-trivial data/algorithms interactions for algorithm selection.

writing) no single method can handle a multitude of shifts (Wiles et al., 2021; Nguyen et al., 2021; Ye et al., 2022; Liang & Zou, 2022; Yang et al., 2023).

> *"It would be helpful for practitioners to be able to select the best approaches without requiring comprehensive evaluations and comparisons."* (Wiles et al., 2021)

**We posit that OOD generalization can be improved if we knew which algorithm to apply in each situation.** However, under complex real-world shifts, finding the right learning algorithm without trial and error is elusive for human experts. We therefore investigate the possibility of *learning* to predict a priori, for a new unseen dataset, the algorithm that will produce the most robust model. This *a priori* prediction contrasts with traditional model selection that first requires training *multiple* models and often relies on restrictive heuristics (Garg et al., 2022; Baek et al., 2023; Miller et al., 2021; Teney et al., 2023; Liu et al., 2024). In a concurrent work, Bell et al. (2024) proposed to select an algorithm to deal with spurious correlations based on past performance on benchmarks most similar to the target data. In comparison, we target many types of shifts (spurious correlation, covariate shift, label shift) and propose a learning-based approach instead of similarity heuristics.

**OOD-CHAMELEON.** We frame the algorithm selection as a classification over a set of candidate algorithms, and study it in a data-driven manner. We train an algorithm selector named OOD-CHAMELEON in a setup akin to *meta-learning* (Vanschoren, 2018; Öztürk et al., 2022) in two ways. (1) It is trained on a *dataset of datasets* that exemplify diverse types, magnitudes, and combinations of shifts. Concretely, we construct such a dataset of datasets by sampling from synthetic distributions or real datasets e.g. CelebA (Liu et al., 2015), in controlled ways to simulate various distribution shifts. (2) We seek to *learn to learn*, or learn to apply a learning algorithm, by discovering interactions between a dataset's characteristics and the performance of candidate algorithms. We evaluate the performance of various algorithms on the dataset of datasets, and we train the algorithm selector with supervision to predict the relative performance of the algorithms given a statistical descriptor of a dataset (see Figure 1). We investigate training objectives such as regression, multi-label classification (Fürnkranz et al., 2008), and pairwise comparison (Bradley & Terry, 1952).

**Our experiments** start with a controllable setup to evaluate various design choices. We then push the concept further with realistic image data. Results show that the system consistently selects algorithms with significantly lower test error than any single candidate algorithm, on unseen datasets with complex types of distribution shifts. We further verify that it achieves this by learning non-trivial, non-linear data/algorithm interactions. More importantly, we show transfer across datasets by training the selector on CelebA-derived datasets, then applying it to COCO (Lin et al., 2014). As a byproduct, we demonstrate that the algorithm selector can reveal which dataset characteristics are important for any candidate algorithm to outperform another.

**In summary**, our findings open new avenues for improving OOD generalization by learning to better apply existing algorithms, instead of designing new ones. Additionally, this line of research can help understand the applicability of existing algorithms, especially under complex conditions such as combined distribution shifts that are difficult to study theoretically.

**Our contributions** are summarized as follows.

- We propose the new task of *algorithm selection for OOD generalization* in a standard setting where attribute labels of the training data are available (Section 2). We present a first proof of concept that predicts the most suitable strategy to obtain a robust model on a given dataset.

- We describe a reusable workflow to construct a *dataset of datasets* that exemplifies a variety of distribution shifts in different types, magnitudes, and combinations. (Section 3.1).

- We propose a demonstrator of the viability of the task (Section 3): an algorithm selector trained with supervised learning to predict the most suitable algorithm from a set of candidates.

- We empirically verify that the proposed model predicts algorithms leading to high OOD accuracy, by learning non-trivial data/algorithm interactions (Section 4). We also show that the model can reveal properties of datasets that make algorithms more effective than others (Section 3.4).

## 2 FORMALIZING THE ALGORITHM SELECTION FOR OOD GENERALIZATION

### 2.1 IS THE SELECTION OF THE BEST LEARNING ALGORITHM EVEN POSSIBLE?

A universal solution to the selection of a learning algorithm cannot exist according to the no-free-lunch theorem (Wolpert & Macready, 1997)). The key here is to restrict ourselves to a *distribution of distribution shifts* that are likely in real-world data (Wiles et al., 2021; Goldblum et al., 2023). We consider the three major types of shifts (Yang et al., 2023): **covariate shifts (CS)**, **label shifts (LS)**, and **spurious correlations (SC)**. Formally, given a joint distribution $P(X, Y)$ over inputs $X$ and labels $Y$, the three types correspond to shifts of $P(X)$, $P(Y)$ and $P(Y|X)$. Each sample $x \in X$ is typically characterized by both a robust feature $x_c \in X_c$ that is reliably predictive, and an attribute $a \in A$ that is not. A covariate shift implies a variation of the attribute $a$ such as a shift in the background typically associated with a specific object in an image. A shift of spurious correlations implies a variation of an attribute/label co-occurrences, which means a shift on $P(Y|A)$ but not $P(Y|X_c)$. Because of the co-occurrence, a model could learn to rely on parts of the input related to the attribute. This model would become unreliable on data where the co-occurrence has shifted. Finally, to measure OOD performance, We use the standard **worst-group accuracy (WGA)** on test data. A **group** $\mathcal{G} \in Y \times A$ refers to a unique attribute–label combination (Sagawa et al., 2020a). We focus on the most common setting in the study of OOD generalization (Yong et al., 2022) where the training data includes labels of a potentially-spurious attribute $a$. We will also evaluate the use of pseudo-attribute labels obtained from a heuristic method in Appendix E.

### 2.2 OOD ALGORITHM SELECTION AS A META-LEARNING TASK

Our eventual goal is to obtain a robust model given a dataset, referred to below as the **OOD task**. An OOD task is defined by its dataset $D = D^{\mathrm{tr}} \cup D^{\mathrm{te}} = \{(x_i, y_i)\}_{i=1}^{n} \cup \{(x_i, y_i)\}_{i=1}^{n_{\mathrm{te}}}$, where training and test data show potential distribution shifts. Solving an OOD task means running a learning algorithm $\mathcal{A}(\cdot) : D^{\mathrm{tr}} \to h_\theta$ that takes the training set and produces a parametrized model. We can then evaluate the performance of $\mathcal{A}$ by computing $\mathrm{WGA}(\{h_\theta(x_i)\}_{i=1}^{n_{\mathrm{te}}}, \{y_i\}_{i=1}^{n_{\mathrm{te}}})$ ($D^{\mathrm{te}}$ is not available during training).

To better solve OOD tasks, we propose to automate the **OOD algorithm selection** (i.e. selecting the best $\mathcal{A}$ among candidates) by learning from experiences over many OOD tasks. The training data $\mathbb{D}$ (a **meta-dataset**) for this process is built from a **dataset of datasets** that represent a variety of distribution shifts. Each example in this meta-dataset $\mathbb{D} = \{D_j^{\mathrm{tr}}, \mathcal{A}_j, P_j\}$ is an OOD task annotated with the performance $P_j$ of an algorithm on this task. In practice, we feed $D^{\mathrm{tr}}$ into a **dataset descriptor** (Rivolli et al., 2022; Jomaa et al., 2021) that summarize its distributional characteristics with a function $f(\cdot) : D^{\mathrm{tr}} \to \mathbb{R}^l$, so that we can leverage vector-based models. This makes $\mathbb{D} = \{f(D_j^{\mathrm{tr}}), \mathcal{A}_j, P_j\}$. Then, we expect to learn from $\mathbb{D}$ an **algorithm selector** $\phi(w, \cdot) : D^{\mathrm{tr}} \to \mathcal{A}$ that takes a training set and predicts the most suitable algorithm[1]. We will describe in Section 3.2 several realizations of the algorithm selector. A straightforward option is a mapping from a dataset descriptor $f(D^{\mathrm{tr}})$ and algorithm identifier $\mathcal{A}$ to its predicted performance, i.e. $\phi(w, \cdot) : f(D^{\mathrm{tr}}) \times$

---

[1]By abuse of notation, $\mathcal{A}$ represents in practice a one-hot identifier of an algorithm.

$\mathcal{A} \to \mathbb{R}$ that is trained by solving a regression objective:

$$\min_{w} \mathbb{E}_{\{f(D_j^{\mathrm{tr}}), \mathcal{A}_j, P_j\} \sim \mathbb{D}} \ \mathcal{L}_O(\phi(w, \{f(D_j^{\mathrm{tr}}), \mathcal{A}_j\}), P_j) \tag{1}$$

where $\mathcal{L}_O$ is a loss function such as the MSE. The trained $\phi(w, \cdot)$ can then be used on any new dataset (representing a new unseen OOD task) to predict the performance of all the candidate algorithms. The best algorithm is eventually used on the new dataset to obtain a robust downstream model. We will show in Section 3.2 that the regression can be reformulated as a classification objective.

Overall, our formulation turns the problem of model selection into a standard supervised task, where training examples represent the past performance of algorithms in various situations. While a theoretical treatment goes beyond the scope of this paper, the generalization and needs for training data could thus be studied with standard results from statistical learning theory (Vapnik, 2000). Our approach is also similar to existing work on the analogous problem of learning to select among pre-trained models (Zhang et al., 2023; Achille et al., 2019; Öztürk et al., 2022) since the selection is also approached in a *data-driven* manner. In our case, the rationale is that a selection purely from existing theory on distribution shifts is intractable because of the complex mixtures of different types of shifts that appear in real-world data.

## 3 OOD-CHAMELEON: LEARNING WHEN TO USE A LEARNING ALGORITHM

We now describe our solution to tackle OOD algorithm selection including *three consecutive steps*:

1. **Obtaining a collection of example datasets** with a variety of distribution shifts to learn from.
2. **Assembling the meta-dataset** $\mathbb{D}$**.** That is, training candidate algorithms on each dataset to get the corresponding performance, and summarizing each dataset with some dataset descriptor $f(\cdot)$.
3. **Training an algorithm selector** on the meta-dataset to learn robust data/algorithm relations.

Each step poses special challenges which we address as follows.

### 3.1 CONSTRUCTING DATASETS WITH VARIOUS DISTRIBUTION SHIFTS

We first generate a dataset of datasets that exhibit three types of shifts in various combinations and degrees. The generation procedure takes in: (1) a triple $(d_{\mathrm{cs}}, d_{\mathrm{ls}}, d_{\mathrm{sc}}) \in [0, 1]^3$ that specifies the degree of covariate shift (CS), label shift (LS), and spurious correlation (SC), and (2) the size of the training set $n$. It computes the required number of samples of each group necessary to achieve these properties. The actual dataset can then be built with these numbers using synthetic data (Section 3.3) or by resampling an existing real dataset such as CelebA (Liu et al., 2015) (Section 4).

**Quantifying distribution shifts.** We need a way to specify the degree of each type of shift. Prior work by Yang et al. (2023) has used information-theoretic measures to quantify the degrees of SC, LS, and CS respectively as the normalized mutual information $I(a; y)$ between labels and attributes, entropy of class $H(y)$, and entropy of attributes $H(a)$. However, we cannot use them to directly compute the required number of samples for each group. Hence we propose an alternative illustrated in Figure 2 with a 2-way classification of shapes with two color attributes. The four groups correspond to $\blacksquare \ \mathcal{G}_1 = \{i | y_i = 1, a_i = 1\}$, $\bullet \ \mathcal{G}_2 = \{i | y_i = -1, a_i = 1\}$, $\bullet \ \mathcal{G}_3 = \{i | y_i = -1, a_i = -1\}$, $\blacksquare \ \mathcal{G}_4 = \{i | y_i = 1, a_i = -1\}$. We define the degree of SC as the ratio of samples where class labels and attribute labels agree, i.e. where a correct classification of the attribute entails a correct classification of the class We define the degrees of LS and CS as the ratio of class/attribute as follows (numerators colored as in Figure 2):

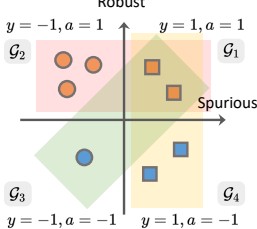

Figure 2: **Illustrative example of our distribution shift quantification.**

$$d_{\mathrm{sc}} = \frac{|\mathcal{G}_1| + |\mathcal{G}_3|}{\sum_i |\mathcal{G}_i|}, \quad d_{\mathrm{ls}} = \frac{|\mathcal{G}_1| + |\mathcal{G}_4|}{\sum_i |\mathcal{G}_i|}, \quad d_{\mathrm{cs}} = \frac{|\mathcal{G}_1| + |\mathcal{G}_2|}{\sum_i |\mathcal{G}_i|}, \tag{2}$$

where $|\cdot|$ is the set cardinality. These degrees are in $[0, 1]$ by definition, and $\sum_i |\mathcal{G}_i| = n$ size of $D^{\mathrm{tr}}$. To construct a dataset with desired shifts, we start with the three desired degrees and the training set

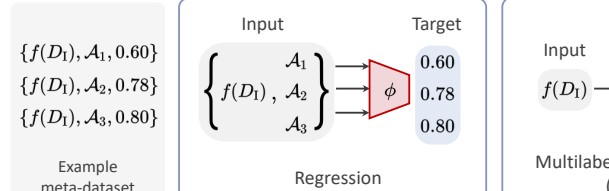
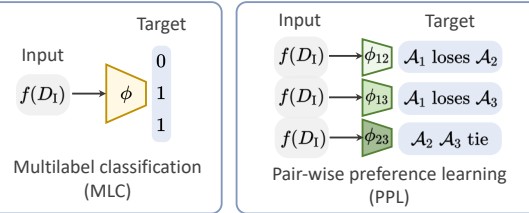

Figure 3: **We propose three possible objectives to train the algorithm selector.** The meta-dataset $\mathbb{D}$ contains here the performance of three algorithms $\mathcal{A}_{\{1,2,3\}}$ on one dataset $D_{\mathrm{I}}$. Regression (**left**) estimates the algorithms' absolute performance. MLC (**middle**) and PPL (**right**) estimate the binarized (unary and pairwise) suitability of the algorithms.

size, then solve linear equations for $|\mathcal{G}_i|$ and obtain the required group sizes in the training set. We sample a balanced test set $D^{\mathrm{te}}$ with $|\mathcal{G}_i| = n_{\mathrm{te}}/4$. See Appendix A for additional explanations.

## 3.2 Formulations of the Algorithm Selector

Given a collection of datasets $\{D_j\}_{j=1}^N$ created as described above, we apply a set of algorithms $\{\mathcal{A}_m\}_{m=1}^M$ on each. Each case yields a downstream model with OOD performance $P_{jm}$ that we store in a meta-dataset $\mathbb{D} = \{f(D_j^{\mathrm{tr}}), \mathcal{A}_m, P_{jm}\}_{j,m}$ (of size $MN$) where $f(\cdot) : D^{\mathrm{tr}} \to \mathbb{R}^l$ is the dataset descriptor (will discuss in Section 3.3). This meta-dataset is used to train the algorithm selector $\phi$ (an MLP in most of our experiments) with one of the three objectives described below The different formulations are visually summarized in Figure 3.

**Formulation 1 (Regression): classification via regression.** Equation 1 tackles the algorithm selection task with a classification via regression. The algorithm selector $\phi(w, \cdot) : f(D^{\mathrm{tr}}) \times \mathcal{A} \to \mathbb{R}$ takes the dataset descriptors and a one-hot algorithm vector as input, predicts the OOD performance, and then chooses the algorithm with the highest one. This may be suboptimal because of (1) the mismatch between the goal of *directly selecting (i.e., classifying)* suitable algorithms, and (2) a regression being often more difficult than classification with neural networks (Devroye et al., 2013). **This motivates two following alternative formulations as classification tasks.**

**Formulation 2 (MLC): multi-label classification.** For each dataset $D_j$, we have $M$ (the number of algorithms) records $\{f(D_j^{\mathrm{tr}}), \mathcal{A}_m, P_{jm}\}_{m=1}^M$ in the meta-dataset $\mathbb{D}$. We aggregate each such set of $M$ records into a *single* training sample $\{f(D_j^{\mathrm{tr}}), Y_\mathcal{A}\}$ where $Y_\mathcal{A} \in \{0,1\}^M$ is a one- or multi-hot vector indicating the suitability of one or several algorithms on $D_j$. An algorithm is considered suitable if $(P_{jm} - \min_m P_{jm}) \leq \epsilon$ for a small threshold $\epsilon$ (e.g. 0.05). This aggregation converts the performance numbers into discrete labels and also serves as "denoising" because we consider algorithms with close performance similarly suitable. The process results in a smaller meta-dataset (from $MN$ to $N$) but allows for learning more robust data/algorithm interactions because of denoising. We train the algorithm selection as a binary multi-label classifier $\phi(w, \cdot) : f(D^{\mathrm{tr}}) \to \{0,1\}^M$ with a cross-entropy objective $\min_w \mathcal{L}(\phi(w, f(D_j^{\mathrm{tr}})), Y_\mathcal{A})$.

**Formulation 3 (PPL): pairwise preference learning.** Going one step further, an alternative formulation decomposes the M-way multi-label classification into $\binom{M}{2}$ classifications (Hüllermeier et al., 2008) across pairs of algorithms. The intuition is that pairwise comparisons are easier than comparing all the candidates (Bradley & Terry, 1952) as recently shown i.a. in the alignment of language models (Ouyang et al., 2022). We train $\binom{M}{2}$ pairwise classifiers $\phi_{km}(w, \cdot) : f(D^{\mathrm{tr}}) \to \{\texttt{win}, \texttt{lose}, \texttt{tie}\}$ to predict the relative ranking of $\mathcal{A}_k$ and $\mathcal{A}_m$. They are considered on par when $|P_{jk} - P_{jm}| \leq \epsilon$. Formally, these 3-way classifiers are trained on $\{f(D_j^{\mathrm{tr}}), Y_\mathcal{A}\}, j \in [1, ..., N]$, where $Y_\mathcal{A} \in \{\texttt{win}, \texttt{lose}, \texttt{tie}\}$. At test time, for a target dataset, we aggregate the $\binom{M}{2}$ predictions using Copeland's classical voting method (Saari & Merlin, 1996). It starts from 0 for all algorithms and adds/subtracts 1 for the winner/loser of each pairwise prediction, and does nothing for ties. The final scores give a ranking across all algorithms from which we select the top value.

| Metrics | Oracle Selection | Random Selection | Global Best | MLC (Naive descriptors) | Regression (Ours) | MLC (Ours) | PPL (Ours) |
|---|---|---|---|---|---|---|---|
| 0–1 ACC. (%) ↑ | 100 | $62.9_{\pm0.6}$ | $72.5_{\pm0.7}$ | $52.1_{\pm0.1}$ | $79.7_{\pm0.7}$ | $\underline{86.3}_{\pm0.4}$ | $\mathbf{90.8}_{\pm0.9}$ |
| Worst-group error (%) ↓ | 19.0 | $24.0_{\pm0.1}$ | $22.7_{\pm0.1}$ | $23.9_{\pm0.2}$ | $20.4_{\pm0.3}$ | $\mathbf{19.9}_{\pm0.1}$ | $\underline{20.1}_{\pm0.1}$ |
| Remarks | Upper bound | Non-parametrized | | Descriptors from Öztürk et al. (2022) | Eq. 1 | multi-label classification | pairwise comparison |

Table 1: **Results on synthetic experiments.** The learned algorithm selector predicts algorithms with low worst-group test error (average over 2,000 unseen datasets and 3 seeds).

### 3.3 DATASET DESCRIPTORS: A CASE STUDY WITH CONTROLLABLE EXPERIMENTS

Now that we focus on algorithm selector's input – the vectorized datasets (i.e., dataset descriptors). The dataset descriptors should summarize properties relevant to the performance of the various algorithms. Recent work (Nagarajan et al., 2021; Hermann et al., 2023; Yang et al., 2024; Chen et al., 2022; Ye et al., 2022; Wang et al., 2024) discovered various properties related to OOD performance that could serve as dataset descriptors. However, many are difficult or impossible to measure before training, or work only under restricted conditions. Other works on learning model selection (Arango et al., 2024; Öztürk et al., 2022) use simple descriptors (e.g., number of training samples, image channels, number of classes, etc.) that clearly cannot predict OOD performance.

We propose two categories of properties to include in dataset descriptors: (1) **distribution shift characteristics** and (2) **data complexity characteristics**. We hypothesize that the former includes the degrees of the distribution shifts ($d_{sc}$, $d_{ls}$, $d_{cs}$ in Equation 2), and the availability[2] $r$ of the spurious correlation (i.e., how easily the model relies on the spurious feature to make predictions). And the latter includes the size of the training set $n$ and the input dimensionality $d$ (e.g. image resolution in vision data).

**Controllable experiments.** We investigate the relevance of the proposed descriptors with a synthetic test case. We consider the binary classification from Figure 2. The input $X$'s distribution follows the modified synthetic example from Sagawa et al. (2020b). The distribution of each group is defined by their input $x = [x_c, x_a] \in \mathbb{R}^{2d}$, with $x_c$ and $x_a$ of dimension $d$ generated from Gaussian distribution:
$$x_c \mid y \sim \mathcal{N}\left(y\mathbf{1}, \sigma_c^2 I_d\right), \quad x_a \mid y \sim \mathcal{N}\left(a\mathbf{1}, \sigma_a^2 I_d\right)$$
The availability of the spurious features is defined as $r = \sigma_c^2/\sigma_a^2$ (more available when higher). Following Section 3.1, we create $N = 7,392$ datasets $D_{1,\dots,N}$ spanning different $d_{sc}, d_{ls}, d_{cs}$, training set sizes $n$, input dimensionality $d$, and availability $r$ (details in Appendix C).

**Experimental setup.** We construct the meta-dataset $\mathbb{D}$ with the created collection of datasets, train the algorithm selector with different objectives on it, and then evaluate how well the algorithm selector predicts suitable algorithms for unseen datasets. Other relevant details (see also Appendix C):

- **Candidate algorithms.** We select 5 algorithms, namely *ERM* (Vapnik, 2000), *Group-DRO* (Sagawa et al., 2020a), *oversampling* the minority groups, *undersampling* the majority groups, and *logits correction* (Nagarajan et al., 2021) (i.e. adjust the prediction logits with a temperature), because: (1) they are shown to perform comparably with others (Nguyen et al., 2021; Gulrajani & Lopez-Paz, 2020; Yang et al., 2023), (2) *do not* require extensive hyperparameter tuning, (3) can handle different distribution shifts (Nguyen et al., 2021) and (4) span different types of approaches, namely regularization-based, reweighting-based, margin-based and standard ERM. Including more algorithms is straightforward.

- **Obtaining OOD performance on the collection of datasets.** We obtain each algorithm's worst-group error $P_{jm}$ on each dataset's test set $D_j^{te}$. We use a linear classifier for training on each dataset because (1) it is sufficient to solve the synthetic example and was also used by Sagawa et al. (2020b), (2) it is relevant to the common setting of linear probing over frozen features (Setlur et al., 2024; You et al., 2024).

- **Evaluation.** We create around 2,000 datasets with unseen properties (i.e. different and unseen dataset descriptors) to test the generalizability of the algorithm selector.

---

[2]Similar concepts exist in prior works: signal/noise ratio (Yang et al., 2024), magnitude (Wang et al., 2024; Joshi et al., 2023), simplicity (Qiu et al., 2024), and spurious/core information ratio (Sagawa et al., 2020b).

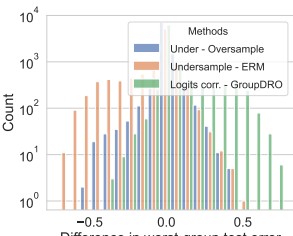 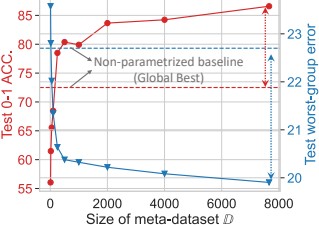 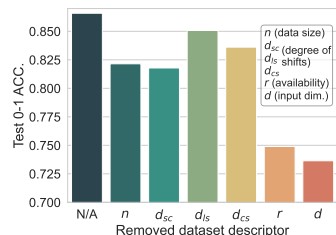

Figure 4: **(Left)** OOD algorithms perform differently across datasets. We compare pairs of algorithms and show the histogram of differences in worst-group test error. **(Middle)** The generalizability of the algorithm selector (MLC) improves with a larger meta-dataset. **(Right)** We estimate the importance of dataset descriptors with leave-one-descriptor-out training of the algorithm selector.

- **Algorithm selection baselines.** (1) *Random selection*: randomly selecting an algorithm per dataset. (2) *Global best (GB)*: choosing the single top algorithm based on its performance over the whole meta-dataset. (3) *MLC (Naive descriptors)*: MLC with the naive dataset descriptors from Öztürk et al. (2022). This is useful to indicate whether our dataset descriptors provide useful additional information. (4) *Oracle selection*: using the ground truth best algorithm per dataset.

**Results.** In Table 1, we show that the learned algorithm selectors generalize to unseen datasets. We examine the 0–1 accuracy, which considers an algorithm prediction on an unseen dataset as correct if it belongs to the "ground truth" suitable algorithms as defined earlier in Section 3.2. We also look at the worst-group error of the selected algorithms, averaged across unseen datasets. We see that (1) our formulations accurately predict suitable OOD algorithms, with significantly higher 0–1 accuracy and lower worst-group error than the baselines. (2) The algorithm selectors generalize significantly better with our dataset descriptors in lieu of naive descriptors, c.f. columns 4 and 6. (3) Comparing our three formulations (columns 4, 5, 6) shows that a classification objective is significantly better than a regression.

In Figure 4 (left), we verify that the different OOD algorithms perform differently across datasets (each count in the histogram is a dataset). This confirms that using the right algorithm for the right dataset improves OOD generalization. In Figure 4 (middle), we show that the algorithm selector (MLC) generalizes better with a larger meta-dataset, yet it already significantly outperforms the best non-parametrized baseline (Global best) even with a small meta-dataset. In Figure 4 (right), we conduct a leave-one-descriptor-out training of MLC by excluding one part of the descriptor at a time. The accuracy drops (c.f. the leftmost bar) reveal the importance of each piece of information for an accurate prediction. The degrees of shifts and data complexity matter most, specifically the input dimensionality $d$, the availability of the spurious feature $r$, and the degree of SC $d_{sc}$.

## 3.4 Attributing Algorithm Effectiveness to Data Characteristics

We take an even closer look at the learned data-algorithm interactions with leave-one-descriptor-out training of **pairwise** algorithm selectors (i.e., selecting from only two algorithms). In Figure 5, for each pair of algorithms, the performance drop of each bar compared to the leftmost bar ("N/A") indicates the significance of the corresponding information for distinguishing the two algorithms. For example, when comparing over- and under-sampling, the data size $n$ and degree of spurious correlation $d_{sc}$ matter most. We find this to be consistent with the analysis in Nguyen et al. (2021) that implies that, while undersampling can cope with more distribution shifts than oversampling, it is inferior when the number of samples for the mi-

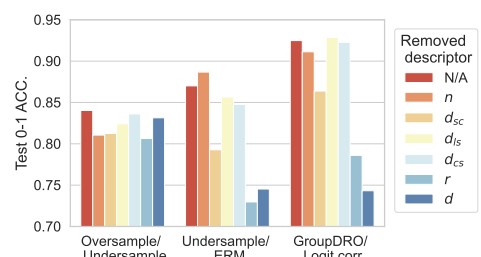

Figure 5: **Leave-one-descriptor-out training of pairwise algorithm selectors.** The accuracy drops reveal decisive factors for one algorithm to outperform another.

nority group is too small (i.e. when $n$ or $d_{sc}$ is too small). This approach can thus help discover data characteristics important for one algorithm to outperform another, and better understand the applica-

| Methods | CelebA | | | | COCO | | | |
| --- | --- | --- | --- | --- | --- | --- | --- | --- |
| | ResNet18 | | CLIP (ViT-B/32) | | ResNet18 | | CLIP (ViT-B/32) | |
| | 0-1 ACC. ↑ | WG error ↓ | 0-1 ACC. ↑ | WG error ↓ | 0-1 ACC. ↑ | WG error ↓ | 0-1 ACC. ↑ | WG error ↓ |
| Oracle Selection | 100 | $40.4_{\pm 0.2}$ | 100 | $31.8_{\pm 0.3}$ | 100 | 33.2 | 100 | 20.9 |
| Random Selection | $25.0_{\pm 1.7}$ | $49.8_{\pm 0.5}$ | $26.7_{\pm 0.9}$ | $41.6_{\pm 0.1}$ | $31.9_{\pm 0.3}$ | $42.0_{\pm 0.2}$ | $38.3_{\pm 0.2}$ | $28.0_{\pm 0.2}$ |
| Global Best (GB) | $44.8_{\pm 1.0}$ | $45.9_{\pm 0.3}$ | $43.0_{\pm 2.2}$ | $38.7_{\pm 0.2}$ | $42.5_{\pm 0.1}$ | $37.3_{\pm 0.3}$ | $46.8_{\pm 0.4}$ | $26.2_{\pm 0.3}$ |
| Regression (Ours) | $\underline{75.4}_{\pm 1.5}$ | $\underline{42.5}_{\pm 0.6}$ | $72.9_{\pm 1.1}$ | $34.3_{\pm 0.3}$ | $51.1_{\pm 1.0}$ | $37.2_{\pm 0.5}$ | $68.8_{\pm 0.6}$ | $24.0_{\pm 0.5}$ |
| MLC (Ours) | $69.1_{\pm 2.1}$ | $42.7_{\pm 0.5}$ | $\underline{80.6}_{\pm 0.4}$ | $\underline{33.5}_{\pm 0.4}$ | $\underline{55.3}_{\pm 0.7}$ | $\underline{36.4}_{\pm 0.3}$ | $\underline{74.4}_{\pm 1.1}$ | $\underline{23.6}_{\pm 0.7}$ |
| PPL (Ours) | $\mathbf{80.0}_{\pm 1.1}$ | $\mathbf{42.0}_{\pm 0.2}$ | $\mathbf{83.7}_{\pm 0.6}$ | $\mathbf{33.0}_{\pm 0.3}$ | $\mathbf{65.9}_{\pm 0.5}$ | $\mathbf{35.9}_{\pm 0.3}$ | $\mathbf{75.8}_{\pm 0.5}$ | $\mathbf{23.4}_{\pm 0.4}$ |

Table 2: **Results on algorithm selection for unseen CelebA and COCO datasets**. Algorithm selectors are trained on the meta-dataset generated from CelebA and evaluated on unseen CelebA or COCO datasets. ResNet18 and CLIP (ViT-B/32) refer to the models use in the OOD tasks.

bility of existing algorithms. We observe for example that the importance of some descriptors varies for other pairs of algorithms (Figure 5 middle/right), which could deserve further investigations.

# 4 Towards Realistic OOD Algorithm Selection

We now push the concept further with realistic data to answer the following research questions. **(RQ1)** Is the learned algorithm selection also effective with datasets of real images? **(RQ2)** Does our algorithm selector, trained on the semi-synthetic datasets built from CelebA for example, transfer to unseen datasets from another domain (e.g. COCO (Lin et al., 2014))? **(RQ3)** How is the system affected by different dataset descriptors and different training variations of OOD tasks (linear probing vs. fine-tuning)? **(RQ4)** How complex are the learned data-algorithm interactions?

**Experimental setup.** The setup resembles that of Section 3.3 (see Appendix D for more details).

- **Dataset of datasets.** We generate 1,056 example datasets from CelebA (Liu et al., 2015) with various sizes, types of shifts, etc. We simulate various "availabilities" of the spurious feature by using various annotations from CelebA, e.g. mouth slightly open as class label and wearing lipstick as spurious attribute (see Table 8).

- **Dataset descriptors.** We now *estimate* the availability of spurious feature (rather than using ground truth values as in Section 3.3) with $r = \sum_y d_y / \sum_a d_a$, where $d_y$ and $d_a$ are average distances of the samples' embeddings to the cluster center of labels or attributes, respectively. Using estimated availabilities allows for generalizing to unseen availabilities, see Appendix D.1 for the rationale and details.

- **Obtaining OOD performance on the collection of datasets.** We use a pre-trained ResNet18 (He et al., 2015) or CLIP model (ViT-B/32) (Radford et al., 2021) to solve each OOD task and obtain the "ground truth" performance of each algorithm. We do a linear probing or a fine-tuning (on ResNet18 for the experiment of Table 5). In addition, we train for long enough (1000 epochs) to ensure convergence with the same hyperparameters in each run. The rationale is that no OOD validation data should be relied on for hyperparameter search, otherwise this OOD data could simply be used as additional training data to achieve OOD generalization.

- **Evaluation**. We create 264, 47 and 150 unseen datasets with CelebA, COCO, and Colored-MNIST respectively. We use the {cat, dog} and {indoor, outdoor} images for COCO. These are classical datasets that we selected because others such as MetaShift (Liang & Zou, 2022) are not large enough to create diverse resampled versions.

**RQ1-RQ2: Algorithm selection is learnable on real-world data.** In Table 2, we see that all three formulations of OOD-CHAMELEON can still select algorithms for unseen OOD datasets from both CelebA and COCO with significantly lower worst-group error and higher 0–1 Accuracy than any baselines. The results on COCO also verify the robustness of our dataset descriptors in capturing general properties relevant to the performance of the algorithms in a way that transfers across datasets. See additional experiments on Colored-MNIST in Appendix F. **The transferability from CelebA to COCO and Colored-MNIST shows the potential of training the algorithm selector once, then using it on other datasets.** Additionally, we see that PPL achieves the best performance among the proposed three realizations. This uses a classification objective with pairwise comparisons (see additional discussion in Appendix B).

| Methods | ResNet18 | | CLIP (ViT-B/32) | |
|---|---|---|---|---|
| | Alg. Selection | WG error ↓ | Alg. Selection | WG error ↓ |
| Oracle Selection | ▬▬▬▬ | 40.4 ±0.2 | ▬▬▬▬ | 31.8 ±0.3 |
| ERM | ▬ | 58.8 ±0.3 | ▬ | 49.9 ±0.4 |
| GroupDRO | ▬ | 46.6 ±0.2 | ▬ | 41.3 ±0.2 |
| Logits Correction | ▬ | 54.1 ±0.5 | ▬ | 42.6 ±0.4 |
| UnderSampling | ▬ | 43.1 ±0.4 | ▬ | 34.6 ±0.2 |
| OverSampling | ▬ | 46.9 ±0.3 | ▬ | 41.5 ±0.1 |
| PPL (Ours) | ▬▬▬▬ | **42.0** ±0.2 | ▬▬▬▬ | **33.0** ±0.3 |

Table 3: **Learning to adaptively use OOD algorithms leads to lower worst-group errors than single algorithms**. The bars are colored by the ratios of algorithms.

| Models | 0-1 ACC. ↑ | WG error ↓ |
|---|---|---|
| Linear (ResNet) | 54.8 ±0.6 | 44.8 ±0.4 |
| k-NN (ResNet) | 62.9 ±0.2 | 43.5 ±0.2 |
| MLP (ResNet) | **69.1** ±2.1 | **42.7** ±0.5 |
| Linear (CLIP) | 50.4 ±0.7 | 36.6 ±0.3 |
| k-NN (CLIP) | 65.3 ±0.2 | 34.8 ±0.1 |
| MLP (CLIP) | **80.6** ±0.4 | **33.5** ±0.4 |

Table 4: **Architecture of algorithm selector.** This shows that the learned data-algorithm interactions are non-trivial (see RQ4).

| Methods | Linear probing | | Fine-tuning | |
|---|---|---|---|---|
| | 0-1 ACC. ↑ | WG error ↓ | 0-1 ACC. ↑ | WG error ↓ |
| Oracle Selection | 100 | 40.4 ±0.2 | 100 | 32.3 ±0.4 |
| Regression (Ours) | 75.4 ±1.5 | 42.5 ±0.6 | 73.6 ±1.8 | 34.5 ±0.3 |
| MLC (Ours) | 69.1 ±2.1 | 42.7 ±0.5 | 67.8 ±1.3 | 34.8 ±0.4 |
| PPL (Ours) | **80.0** ±1.1 | **42.0** ±0.2 | **79.2** ±1.3 | **34.1** ±0.3 |

Table 5: **Evaluation of training paradigms for OOD tasks**. The model generalizes on both linear probing and fine-tuning.

| Methods | ResNet18 | | CLIP (ViT-B/32) | |
|---|---|---|---|---|
| | 0-1 ACC. ↑ | WG error ↓ | 0-1 ACC. ↑ | WG error ↓ |
| Regression (*) | 66.3 ±1.1 | 46.3 ±0.2 | 53.4 ±0.9 | 36.8 ±0.1 |
| MLC (*) | 54.9 ±1.3 | 45.0 ±0.3 | 49.8 ±0.9 | 37.2 ±0.5 |
| PPL (*) | 69.3 ±1.0 | 44.1 ±0.3 | 69.4 ±0.8 | 35.9 ±0.3 |
| Regression (Ours) | 75.4 ±1.5 | 42.5 ±0.6 | 72.9 ±1.1 | 34.3 ±0.3 |
| MLC (Ours) | 69.1 ±2.1 | 42.7 ±0.5 | 80.6 ±0.4 | 33.5 ±0.4 |
| PPL (Ours) | **80.0** ±1.1 | **42.0** ±0.2 | **83.7** ±0.6 | **33.0** ±0.3 |

Table 6: **Comparison of our dataset descriptors** with the simple ones (marked *) from Öztürk et al. (2022).

In Table 3, we compare PPL with single-algorithm baselines, where a single algorithm is used for all unseen datasets. Our method performs an *adaptive* selection of algorithms for each unseen dataset, thereby achieving lower worst-group error than any single algorithm. Furthermore, the ratios of the selected algorithms across unseen datasets are close to the ground truth oracle selection. This also shows that choosing algorithms according to the nature of the dataset is both learnable and helpful.

**RQ3: Ablation studies.** We analyze the impact of dataset descriptors and training paradigms for the OOD tasks. In Table 6, we show that the simple dataset descriptors from Öztürk et al. (2022) are clearly outperformed by ours. In Table 5, we verify that the algorithm selector works well with both linear probing and fine-tuning with CelebA. This is necessary to check because, solving the OOD tasks by different training paradigms affects the algorithms' OOD performance $P_{jm}$ and therefore changes the distribution of meta-dataset $\mathbb{D} = \{f(D_j^{\text{tr}}), \mathcal{A}_m, P_{jm}\}$. The results indicate that the learned algorithm selector can also accurately select suitable algorithms when the models for the OOD tasks become over-parametrized in the case of fine-tuning.

Importantly, in Appendix E we evaluate our model with *estimated* dataset descriptors, i.e. when information such as the samples' attributes are not directly available at test time. The results show that suitable algorithms are predicted even with the estimated dataset descriptors.

**RQ4: The algorithm selector learns non-trivial data-algorithm interactions.** We evaluate the complexity of the learned data-algorithm interactions by comparing various architectures for the algorithm selector $\phi(\cdot, w)$ (see Table 4; "Linear (ResNet)" means for example a linear model for the selector and a ResNet18 for the OOD task). First, we see that a linear model is significantly worse than an MLP. This shows that the model **makes accurate predictions on the basis of non-linear interactions between datasets' characteristics and algorithms' performance**. Second, we see that a k-NN is also significantly worse than an MLP. This shows that the model **works not only by memorizing a large number of example shifts**, which a k-NN could also do. On the contrary, accurate predictions on unseen datasets require non-trivial generalization.

## 5 RELATED WORK

**OOD generalization** has been widely studied but several benchmark studies concluded that there is no one-fits-all solution to distribution shifts (Gulrajani & Lopez-Paz, 2020; Wiles et al., 2021; Nguyen et al., 2021; Ye et al., 2022; Liang & Zou, 2022; Yang et al., 2023; Bell et al., 2024). These meta-studies show that OOD algorithms perform differently in different situations. An algorithm

only shines when its underlying assumptions are met. This is the motivation for our goal of *adaptively* using existing algorithms, and investigating whether this can be learned and automated.

**Algorithm selection.** Model selection (Forster, 2000; Raschka, 2020) and algorithm selection (Rice, 1976; Kerschke et al., 2019) are integral parts of any machine learning workflow. Regarding OOD generalization, a series of works (Liu et al., 2024; Baek et al., 2023; Garg et al., 2022; Miller et al., 2021; Lu et al., 2023) use heuristics to estimate a models' OOD performance on a target dataset when no labeled OOD data is available. These heuristics have one or several of these downsides: (1) they can only be estimated after training, (2) they require (unlabeled) test data, and (3) they work only under restricted conditions. In contrast, our work aims for *a priori* selection of the best algorithm, i.e. before training on the target dataset.

Concurrently to our work, Bell et al. (2024) identified a very similar motivation. They proposed to select among algorithms that deal with spurious correlations based on their performance on benchmarks most similar to the target dataset. The differences with our work are: (1) we use a *learning* approach rather than a similarity heuristic, (2) our learning process relies on *semi-synthetic* data rather than existing benchmarks, which helps cover a broader set of distribution shifts, and (3) we also consider *label and covariate shifts* and their combinations, not only spurious correlations.

**Meta-learning.** Our "learning-to-learn" process resembles meta-learning (Vilalta & Drissi, 2002; Hospedales et al., 2022) in that it carries over prior experience from historical tasks to future similar ones. Achille et al. (2019); Öztürk et al. (2022); Arango et al. (2024); Zhang et al. (2023) learn to select among pretrained models for downstream tasks or for outlier detection Zhao et al. (2021). They either first require training a model on the target dataset or they are not suited to OOD generalization.

**AutoML.** Our work relates to AutoML which aims to identify ML workflows, often by trial-and-error (Hutter et al., 2019; He et al., 2021). Our work differs in its aim for *a priori* algorithm selection and its potential for unveiling datasets' properties predictive of the applicability of algorithms.

## 6 DISCUSSION

This work explored a new avenue for improving OOD generalization by better using existing algorithms instead of creating new ones. We formalized the task of *OOD algorithm selection* and took the first step to learn it. We treated it as a classification over candidate algorithms, learned with supervision from a *dataset of datasets* representing a variety of distribution shifts. The resulting system not only predicts low-test-error algorithms on unseen datasets, but also reveals key properties of datasets that allow algorithms to outperform one another. More could be learned in the future about existing algorithms e.g. by training the selector as an interpretable decision tree.

**Limitations and future work.** The proposed solution served to verify the viability of the new task. Future work will expand this approach in several ways.

- **Candidate algorithms.** Our proof of concept uses simple algorithms representative of several broad categories. They were selected because they are effective in a range of settings and sometimes even superior to sophisticated alternatives Idrissi et al. (2022). The fact that our approach is effective with such simple algorithms is testament to its potential for better exploiting the plethora of other existing algorithms. A direct future extension will include more algorithms and their variations, e.g. by including some of their hyperparameters in the search space.

- **Can we transfer the approach to real-world settings?** As discussed in Section 2, we consider a restricted but representative distribution of distribution shifts and learn the algorithm selector. To verify that most real-world scenarios fall into this distribution, the approach will be applied to real-world data (e.g. from WILDS, Koh et al. (2021)) and larger-scale models. This will evaluate the limits of transferability of the learned selector as started in Section 4.

- **Are the proposed dataset descriptors optimal?** The proposed dataset descriptors (Section 3.3) are interpretable but (1) they may not capture all relevant data properties, (2) and they require the knowledge of the spurious attribute (as many OOD generalization works do (Yong et al., 2022)). A promising direction is to replace our dataset descriptors with learned representations of datasets e.g. with Set Transformers Lee et al. (2018) or Dataset2Vec Jomaa et al. (2021).

## REPRODUCIBILITY STATEMENT

In order to ensure that this work is reproducible, we have relied on the open-source code of SubpopBench (Yang et al., 2023) for some of the experiments in Section 4 and we have provided the anonymized source code for other experiments performed in the paper.

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

# APPENDIX

We provide more details and results omitted in the main paper, summarized as follows.

- Appendix A details **how we construct datasets with desired distribution shifts**.

- Appendix B provides **intuitions why the 3 formulations of the selector perform differently**.

- Appendix C describes **details on the setup of the controllable experiments** from Section 3.3.

- Appendix D describes **details on the setup of the realistic experiments** from Section 4.

- Appendix E evaluates the use of **estimated dataset descriptors** as input to the selector.

## A  META-DATASET CONSTRUCTION

In Section 3.1, we describe a framework that allows for constructing datasets with diverse distribution shifts by sampling from synthetic distributions or an existing dataset, here we provide more details and examples on this front. There are two use cases in Section 3.3 and Section 4 respectively, where in both cases we know the distribution of each group (recall that the combinations of different values of attribute $a$ and class $y$ form different groups). Specifically, in Section 3.3, we have the group distributions as Gaussian distributions so that we can sample the desired numbers of samples from those distributions, while in Section 4 we have a decent amount of samples (in CelebA) from each group distributions and we can also sample the desired numbers of samples from each group distributions.

In Equation 2, we define the degrees of distribution shifts as a function of the number of samples for each group. Therefore, to obtain a dataset with specific degrees of distribution shifts, one only needs to solve the number of samples for each group and sample them from the group distribution. Note that the number of samples can be scaled up or down depending on the size of the dataset we want. All of the constraints to be solved are therefore:

$$d_{sc} = \frac{|\mathcal{G}_1| + |\mathcal{G}_4|}{\sum_i |\mathcal{G}_i|}, \quad d_{ls} = \frac{|\mathcal{G}_1| + |\mathcal{G}_3|}{\sum_i |\mathcal{G}_i|}, \quad d_{cs} = \frac{|\mathcal{G}_1| + |\mathcal{G}_2|}{\sum_i |\mathcal{G}_i|},$$
$$\sum_i |\mathcal{G}_i| = n, \quad (|\mathcal{G}_i| \geq 0) \tag{3}$$
$$0 \leq d_{sc} \leq 1, \quad 0 \leq d_{ls} \leq 1, \quad 0 \leq d_{cs} \leq 1,$$

Solving the constraints gives the solution set of the degrees of distribution shifts, as shown in Figure 6. We see that not any value in the cube can be chosen because of the constraint $|\mathcal{G}_i| \geq 0$ for $\forall i$.

We can then generate various datasets by varying: (1) the degrees of distribution shifts, (2) the size of the dataset, and (3) the group distributions. Note that by definition when given a $d_{(\cdot)} = 0.5$, it means the absence of the corresponding type of distribution shift. This framework can therefore generate combinations of shifts in types and degrees.

## B  DISCUSSIONS ON THE ALGORITHM SELECTOR

While achieving the same goal of predicting the suitable algorithms given the descriptors of the target dataset, the three formulations of the algorithm sector mentioned in Section 3.2 perform differently as verified in Section 3.3 and Section 4. Here we briefly discuss our intuition on this front.

The disadvantage of regression is that, it is hard to learn robust information from the subtle differences in performance. These subtle differences in performance between algorithms can be caused by the variances of training randomness. Therefore, a regression model trained on continuous performance can be disturbed by the variances of OOD task training.

Transforming the continuous performance into discrete labels and use classification models (what we called a 'denoising' step) is the key to learning robust information from the meta-dataset. Benefiting from this step, the multi-label classification (MLC) trains an end-to-end classifier to take the dataset

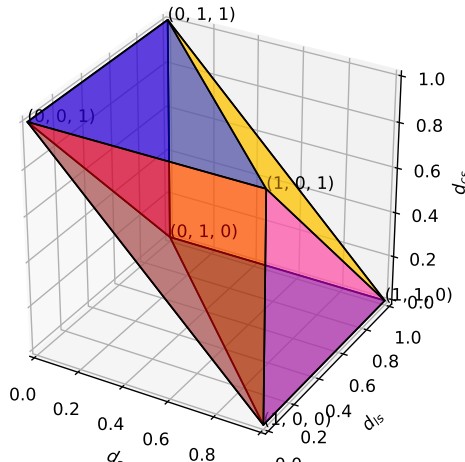

Figure 6: The feasible degrees of spurious correlation ($d_{\text{sc}}$), covariate shift ($d_{\text{cs}}$) and label shift ($d_{\text{ls}}$).

descriptors as input and predict which algorithm is 'suitable' on the target dataset. It is verified to perform significantly better than the regression most of the time.

Finally, leveraging pairwise comparison (PPL) is observed to outperform MLC in many cases. While a future analysis on this front is needed, our hypotheses are that (1) pairwise comparison is easier than comparing all the algorithms at once, which is consistent with our observation that higher accuracies are achieved on each pairwise classifiers (many achieved above 90% and some were close to 100% perfect accuracy), and (2) pairwise classifiers can parameterize more nuanced data-algorithm interactions, while MLC as a single classifier, only learns a more global view. This becomes more clear when comparing Figure 4-right and Figure 5, we see that there exist different 'patterns' of data-algorithm interactions (exemplified by the importance of dataset descriptors) in different pairwise classifiers.

## C    DETAILED SETUP OF THE CONTROLLABLE EXPERIMENTS

We provide the detailed experimental setup for the controllable experiments in Section 3.3.

**Dataset of datasets.**    In total, we created 9,240 datasets, see Table 7 for the statistics for these datasets. Specifically, we generate the training sets of the datasets with the combinations of values in Table 7, and for the degrees of distribution shifts, we consider 2 cases: (1) 3 shifts (spurious correlation, covariate shift and label shift) all present each in different degrees, we uniformly sampled 30 different triple degrees, and (3) there is only 1 shift present, in this case for each shift we 9 different values shown in the table (therefore in total $3*9-2 = 25$ different triple degrees). We generate their test sets with the same number of samples as its training set for each dataset, but keep the number of samples the same for all groups (therefore balanced test sets with all $d_{(.)} = 0.5$). We randomly split these datasets into 4:1 as the dataset of datasets for training the algorithm selector, and the unseen datasets for evaluation.

**Algorithm selector.**    In all formulations of the algorithm selector mentioned in Section 3.2, we use 4-layer MLPs to parameterize the algorithm selector, because we found a shallower or simpler model underfits (see Table 4) while a deeper model does not provide more improvements.

**OOD tasks.**    To solve the OOD tasks defined by each dataset in the dataset of datasets, we use Adam optimizer with default hyperparameters, along with l2 regularization. We train for 1000 epochs to ensure convergence on this synthetic example.

| Statistics | Value |
|---|---|
| Size of training set | $\{200, 500, 1000, 2000, 3000, 5000, 10000\}$ |
| Input dimensionality | $\{2, 10, 50, 100\}$ |
| Availability $r = \sigma_c^2/\sigma_a^2$ | $\{1, 5, 10, 20, 50, 100\}$ |
| 3 shifts | uniformly sampled 30 feasible degrees (Figure 6) |
| 1 shift | $\{0.01, 0.05, 0.1, 0.3, 0.5, 0.7, 0.9, 0.95, 0.99\}$ |

Table 7: Statistics for the dataset of datasets in controllable synthetic experiments.

## D  DETAILED SETUP OF THE REALISTIC EXPERIMENTS

### D.1  AVAILABILITY OF THE SPURIOUS FEATURES

In the controllable experiments in Section 3.3, we compute the availability of spurious features as $r = \sigma_c^2/\sigma_a^2$, following the spurious-core information ratio defined in Sagawa et al. (2020b). The higher the $r$, the more signal there is about the spurious attribute in the spurious features, relative to the signal about the label in the core features.

However, in real-world data (Section 4), it is *not* straightforward how to generate the dataset of datasets with diverse $r$. To tackle this, we use different attribute pairs (one as the label attribute, i.e. the attribute we want to classify, and the other as spurious attribute) as shown in Table 8, to account for different availability of spurious features. Intuitively, the obviousness of different types of attributes (in terms of size, color, etc) signifies different availability of the spurious feature (caused by the attribute).

| Label attribute | Spurious attribute |
|---|---|
| Mouth Slightly Open | Wearing Lipstick |
| Attractive | Smiling |
| Black Hair | Male |
| Oval Face | High Cheekbones |

Table 8: Attribute pairs in CelebA (Liu et al., 2015) that are used to construct different availability of the spurious feature.

By considering different attribute pairs, we are able to generate datasets with different availabilities according to the same framework in Section 3.1. However, unlike the controllable experiments, here we cannot generalize to unseen availability if we directly encode the availability as a one-hot vector indicating the attribute pairs. Instead, as mentioned in Section 4, we compute the availability as the average distances of the samples' embeddings to their labels or attributes clustering center, respectively. Specifically, we first obtain the samples embeddings $\{z_i\}_{i=1}^{n_{tr}}$ from the used backbone (either ResNet18 or CLIP) and then compute the clustering center of each label and attribute as $\boldsymbol{\mu}_y$ and $\boldsymbol{\mu}_a$. The availability $r$ is then:

$$
\begin{aligned}
r &= \frac{\sum_y d_y}{\sum_a d_a} \\
d_y &= ||z_i - \boldsymbol{\mu}_y||_2 \\
d_a &= ||z_i - \boldsymbol{\mu}_a||_2 \\
\boldsymbol{\mu}_y &= \frac{1}{|y_i = y|} \sum_{y_i=y} z_i \\
\boldsymbol{\mu}_a &= \frac{1}{|a_i = a|} \sum_{a_i=a} z_i
\end{aligned}
\tag{4}
$$

This definition is intuitively similar to what is defined for availability in controllable experiments, where we had $r = \sigma_c^2/\sigma_a^2$. Intuitively, a smaller average distance w.r.t. attribute labels signifies an easier spurious feature and therefore, higher availability. This allows for training on datasets with

diverse availabilities (constructed by different attribute pairs) and generalizes to unseen availabilities (e.g. on the COCO datasets).

## D.2 DETAILED EXPERIMENTAL SETUP

We provide the detailed experimental setup for the realistic experiments in Section 4.

**Dataset of datasets.** In total, we created 1,320 datasets from CelebA (Liu et al., 2015), see Table 9 for the statistics for these datasets. Specifically, we generate the training sets of the datasets with the combinations of values in Table 9, and for the degrees of distribution shifts, we consider 2 cases: (1) 3 shifts (spurious correlation, covariate shift and label shift) all present each in different degrees, we uniformly sampled 30 different triple degrees, and (3) there is only 1 shift present, in this case for each shift we 9 different values shown in the table (therefore in total $3 * 9 - 2 = 25$ different triple degrees). We generate their test sets with the half number of samples as its training set for each dataset, but keep the number of samples the same for all groups (therefore balanced test sets with all $d_{(\cdot)} = 0.5$). We randomly split these datasets into 4:1 as the dataset of datasets for training the algorithm selector, and the unseen datasets for evaluation. Additionally, we create 47 datasets from COCO (Lin et al., 2014) for evaluation, where we follow a similar generation strategy. However, we were only able to generate a limited number of datasets since the cat, dog and indoor, outdoor are limited. A future evaluation on datasets that support generating a larger number of datasets is expected.

| Statistics | Value |
| --- | --- |
| Size of training set | {200, 500, 1000, 2000, 5000, 10000} |
| Input dimensionality | N/A |
| Availability | 4 different attribute pairs, see Table 8 |
| 3 shifts | uniformly sampled 30 feasible degrees (Figure 6) |
| 1 shift | {0.01, 0.05, 0.1, 0.3, 0.5, 0.7, 0.9, 0.95, 0.99} |

Table 9: Statistics for the dataset of datasets in realistic experiments.

**Algorithm selector.** In all formulations of the algorithm selector mentioned in Section 3.2, we use 4-layer MLPs to parameterize the algorithm selector, because we found a shallower or simpler model underfits (see Table 4) while a deeper model does not provide more improvements.

**OOD tasks.** To solve the OOD tasks defined by each dataset in the dataset of datasets, we either do (1) linear probing on ResNet18 or CLIP (ViT-B/32) or (2) fine-tuning ResNet18. In the first case (all tables except for Table 5), we use Adam optimizer with default hyperparameters, along with l2 regularization. We train for 1000 epochs to ensure convergence. In the second case (Table 5), we use SubpopBench (Yang et al., 2023) and its default hyperparameters to fine-tune ResNet18. We use basic data augmentations (resize, crop, ...).

## E ALGORITHM SELECTION WITH ESTIMATED DATASET DESCRIPTORS

Here we study the scenarios where the information to compute dataset descriptors, such as the attribute for each sample (Liu et al., 2021), is not easily obtained at test time. Recent works (Liu et al., 2021; Kirichenko et al., 2022; Qiu et al., 2023; Lee et al., 2023; Pagliardini et al., 2022) for OOD generalization aim to eliminate the need for attribute annotation by either: (1) infer the attribute annotation and then use them to run algorithms that require attribute annotation, (2) run ERM on the training set assuming that the ERM learns the spurious feature, and then build invariant classifier on top of the ERM classifier (e.g. fit a model that disagrees with the ERM).

We leverage the above first line of research, i.e. inferring the attribute annotation and use them to compute the dataset descriptors on the target training set. Then, we can use OOD-CHAMELEON to select the suitable algorithms. We infer the attributes by clustering the embeddings from frozen backbones, following Sohoni et al. (2020); You et al. (2024). The intuition is that different attributes,

such as cows in grass or desert, can be considered as 'subclasses' or 'hidden stratifications', and they are observed to be separable in the feature space of the deep models. Hence, for example we can infer which cow belongs to which environment, by clustering on the embeddings. In particular, the training samples are passed through the backbone of the OOD task (i.e. ResNet18 or CLIP in our case), and get the embeddings. For each semantic class, we cluster the corresponding samples with K-means and assign different attribute labels to different clusters. This gives each sample its inferred attribute annotation. We can then use these inferred attribute annotations to compute dataset descriptors, where they are used to compute the degree of spurious correlation $d_{sc}$ and covariate shift $d_{cs}$, as well as the availability of spurious features $r$, see Appendix D.1 on how we compute the availability.

With the inferred dataset descriptors, we use the algorithm selector to predict suitable algorithms. In Tab 10, we show that the suitable algorithms are still predictable with estimated dataset descriptors. Interestingly, while having performance drops in most cases, using estimated dataset descriptors boosts the performance on COCO with ResNet18.

| Methods | Attribute | CelebA | | COCO | |
|---|---|---|---|---|---|
| | | ResNet18 | CLIP (ViT-B/32) | ResNet18 | CLIP (ViT-B/32) |
| Oracle Selection | N/A | 100 | 100 | 100 | 100 |
| Regression | ✓ | $75.4_{\pm1.5}$ | $72.9_{\pm1.1}$ | $51.1_{\pm1.0}$ | $68.8_{\pm0.6}$ |
| MLC | ✓ | $69.1_{\pm2.1}$ | $80.6_{\pm0.4}$ | $55.3_{\pm0.7}$ | $74.4_{\pm1.1}$ |
| PPL | ✓ | $80.0_{\pm1.1}$ | $83.7_{\pm0.6}$ | $65.9_{\pm0.5}$ | $75.8_{\pm0.5}$ |
| Regression | ✗ | $70.8_{\pm1.6}$ | $66.3_{\pm1.4}$ | $55.3_{\pm0.8}$ | $61.9_{\pm1.0}$ |
| MLC | ✗ | $68.2_{\pm0.7}$ | $72.0_{\pm1.1}$ | $61.7_{\pm1.2}$ | $70.2_{\pm0.9}$ |
| PPL | ✗ | $76.5_{\pm0.8}$ | $81.8_{\pm1.0}$ | $71.2_{\pm1.6}$ | $70.4_{\pm0.5}$ |

Table 10: OOD-CHAMELEON with inferred attribute labels (and dataset descriptors) on CelebA and COCO. 0-1 Accuracy (higher is better) is shown.

# F    ADDITIONAL EXPERIMENTS

In Section 4, we investigate with RQ1 and RQ2 the effectiveness of OOD-CHAMELEON in selecting algorithms on real-world data. Furthermore, we show the learned data-algorithm relation is transferrable: when training the algorithm selector with a meta-dataset constructed from CelebA, the algorithm selector proves to be effective on unseen COCO datasets as well.

Here we provide more experiments as further support, in particular, we train on the same CelebA meta-dataset and evaluate the algorithm selector on Colored-MNIST dataset (Arjovsky et al., 2020). In Colored-MNIST dataset, there are images of 10 digits from 0-9 and the digits are divided into 2 classes (i.e. 0–4 is class 0, 5–9 is class 1). In addition, the two classes of digits are in two different colors (e.g. red and green). When the colors of the digits correlate with the shapes of digits, a spurious correlation occurs. Similar to Section 3.1, we create 150 datasets from Colored-MNIST dataset, each dataset exhibits different magnitudes of spurious correlation (SC), label shift (LS) and covaraite shifts (CS) and the size of datasets span {200, 500, 1000, 2000, 5000}. In Table 11, we see that the algorithm selector proves to be effective on Colored-MNIST as well.

| Methods | Colored-MNIST | | | |
| | ResNet18 | | CLIP (ViT-B/32) | |
| | 0-1 ACC. ↑ | WG error ↓ | 0-1 ACC. ↑ | WG error ↓ |
|---|---|---|---|---|
| Oracle Selection | 100 | 21.1 $_{\pm 0.3}$ | 100 | 13.3 $_{\pm 0.4}$ |
| Random Selection | 29.3 $_{\pm 1.1}$ | 28.1 $_{\pm 0.3}$ | 39.3 $_{\pm 0.5}$ | 19.0 $_{\pm 0.3}$ |
| Global Best (GB) | 50.1 $_{\pm 0.7}$ | 25.1 $_{\pm 0.4}$ | 63.3 $_{\pm 1.2}$ | 16.0 $_{\pm 0.2}$ |
| Regression (Ours) | 79.4 $_{\pm 0.5}$ | 23.8 $_{\pm 0.3}$ | 54.2 $_{\pm 0.9}$ | 16.1 $_{\pm 0.4}$ |
| MLC (Ours) | 82.7 $_{\pm 0.8}$ | 23.5 $_{\pm 0.4}$ | 75.3 $_{\pm 0.7}$ | 15.6 $_{\pm 0.4}$ |
| PPL (Ours) | **82.8** $_{\pm 0.4}$ | **23.3** $_{\pm 0.2}$ | **85.3** $_{\pm 0.6}$ | **15.3** $_{\pm 0.3}$ |

Table 11: **Results on algorithm selection for unseen Colored-MNIST datasets**. Algorithm selectors are trained on the meta-dataset generated from CelebA and evaluated on unseen Colored-MNIST datasets. ResNet18 and CLIP (ViT-B/32) refer to the models use in the OOD tasks.

