# OpenReview forum: "OOD-Chameleon: Is Algorithm Selection for OOD Generalization Learnable?"
_ICLR.cc/2025/Conference — Submitted to ICLR 2025_

### Official Review · Reviewer_5F9j · 2024-10-24

**Soundness:** 2
**Presentation:** 2
**Contribution:** 2
**Rating:** 3
**Confidence:** 3

**Summary:**

This work introduces OOD-CHAMELEON, a method for selecting the most suitable learning algorithm for out-of-distribution (OOD) generalization challenges. By treating algorithm selection as a supervised classification problem, the proposed solution learns from a dataset of diverse shifts to predict the relative performance of algorithms based on a dataset’s characteristics. This allows for the a priori selection of the best learning strategy. Experimental results demonstrate that the adaptive selection approach outperforms individual algorithms and simple heuristics on unseen datasets with controllable and realistic image data, revealing some interactions between data and algorithms.

**Strengths:**

1) The approach improves the ability to generalize in OOD scenarios by selecting the most appropriate algorithm for a given dataset’s characteristics.
2) OOD-CHAMELEON eliminates the need for training and evaluating multiple models for algorithm selection, leading to a more efficient use of computational resources.
3) The method provides interesting insights into the conditions under which different algorithms are more suitable.

**Weaknesses:**

1) The underlying concept of this article seems to require theoretical support. Selecting the most appropriate model based on data characteristics appears to be more challenging than learning a predictive model. To truly excel in this area, I believe a substantial number of datasets are needed for validation. However, in real-world scenarios, there may not be an abundance of datasets available. Therefore, further analysis and discussion are needed to determine the minimum number of datasets required to effectively choose the right model.
2) The experimental section of this paper uses a few datasets and models, which is insufficient to fully validate the method proposed in this paper and the insights provided.
3) The performance of a model also depends on how well it is optimized on the dataset. Please provide specific details on the training process for each model, particularly whether the model’s parameters have been optimized to their best possible values.

**Questions:**

Directly training multiple models and then aggregating their outputs might yield better results than the method proposed in this paper. In practical applications, what are the advantages of the method proposed in this paper compared to the ensemble model approach?

**Details Of Ethics Concerns:**

nan

---

> ### Author Response · Authors · 2024-11-20
> **Response to Reviewer 5F9j**
>
> We thank the reviewer for their time and effort, and for acknowledging the advantages and new insights of our work.
>
> > **W1**. Requires more theoretical support.
>
> The whole point of this paper is the possibility of going beyond existing theoretical characterizations of shifts and of assumptions of OOD algorithms. We will make this clearer in the paper. Many theoretical results have limited real-world applicability because the data often contains **mixtures of different types of shifts**, which cannot be easily characterized by theory. Our meta-learning approach turns algorithm selection into a supervised problem where classical results from statistical learning theory apply.
>
> We also refer to prior work [1-4] (Section 5) that also used dataset characteristics as input for model selection with an empirical approach. The goal was then selecting among pretrained models, rather than OOD algorithms.
>
> [1] Quick-Tune: Quickly Learning Which Pretrained Model to Finetune and How. Arango et al. ICLR 2024
> [2] Zero-shot AutoML with Pretrained Models. Öztürk et al. ICML 2022
> [3] TASK2VEC: Task Embedding for Meta-Learning Achille et al. ICCV 2019
> [4] Model Spider: Learning to Rank Pre-Trained Models Efficiently Zhang et al. NeurIPS 2023
>
> > **W2**. Few datasets.
>
> The CelebA and COCO were deemed the most suitable for the controlled, yet realistic experiments that would support our claims. As mentioned at L409, other datasets such as MetaShift have an insufficient number of samples per each group for creating resampled versions. We welcome suggestions from the reviewer and we will be happy to include additional evaluations in the final version of the paper.
>
> > **W3**. The performance of a model also depends on how well it is optimized on the dataset. Please provide specific details on the training process for each model, particularly whether the model’s parameters have been optimized.
>
> Most of the algorithms we chose do not have multiple hyperparameters to be tuned, unlike other OOD generalization algorithms. Only GroupDRO has a hyperparameter and we keep it the default value as suggested by the original paper. For each algorithm, we keep the learning rate and batch size the same value after searching over a smaller set of datasets, and we run the algorithm long enough so that the convergence is ensured.
>
> This whole procedure is valid because in practice one should not expect a sufficient number of OOD validation samples for hyperparameter search, otherwise why not just bypass OOD generalization by training on OOD validation samples? We will clarify in the paper, thanks for pointing out.
>
> > **Q1**. Advantages over ensemble?
>
> Reasons against ensembles:
>
> - **Inefficient**. This requires training and evaluating multiple models. Our approach predicts a single algorithm to train a single model on the target dataset.
>
> - **Do not address distribution shifts** unless the majority of the models already make correct predictions. If most of the models used for output aggregation suffer from the same problem (e.g. a spurious correlation), aggregating their outputs does not help.

---

> ### Comment · Reviewer_5F9j · 2024-11-27
>
> Thank you for your reply. However, I believe that for a purely experimental paper, more datasets (like iwilds, nico++), model types (like ensemble approach), and comprehensive testing are needed to reflect the paper's contribution. Therefore, I am maintaining the original score.

---

> > ### Author Response · Authors · 2024-11-29
> >
> > We sincerely appreciate the reviewer's consideration. We highlight one last time that opening up this new avenue for tackling OOD generalization already includes substantial technical contributions:
> >
> > - the formalization of the new task and an evaluation of its feasibility under controlled conditions;
> > - the design and evaluation of several training objectives;
> > - the design and evaluation of meta-dataset generation strategies;
> > - the evaluation of a learned selector transfer across datasets.
> >
> > The scale-up will certainly bring additional challenges worthy of entire future publications.

---

### Official Review · Reviewer_bNot · 2024-11-01

**Soundness:** 3
**Presentation:** 4
**Contribution:** 3
**Rating:** 6
**Confidence:** 3

**Summary:**

This work focuses on the development of a method for selecting algorithms, specifically with the goal of OOD generalization. Optimal selection of algorithms may depend on the type of shift which has occurred and what the algorithm is tailored to handle.


The proposed selection method, named OOD-chameleon, is a learnable method which aims to solve a classification task over a set of candidate algorithms.

The supervision of this task is proposed to be based on a meta-dataset which contains many datasets with different types of shifts and the candidate algorithms performances on these. The datasets are represented by dataset descriptors which contain measures of the distribution shift and data complexity.

These datasets are then constructed by sampling from synthetic distributions or from real-world datasets in the empirical evaluation of the method. The evaluation shows that the method selection performs better than just selecting the best performing model overall. Further, the authors show that leaving some dataset descriptors out of the meta dataset description can severely hamper the performance of the selected algorithm. This implies that certain information is more valuable when making a selection, depending on the algorithm characteristics.

**Strengths:**

- The possibility of learning how to select algorithms based on dataset characteristics is very interesting

- The motivation for the work is clear

- The performance of the method is good and makes the case for using properties of the data to select different algorithms on a per dataset  basis

**Weaknesses:**

- My main concern is the need to know the attributes that are shifting a priori. Finding the proposed metrics for different real-world datasets is not something straightforward. It is furthermore unclear how we would get at these metrics in general if they are not given.

- The choice of attributes seem central for the approach to work at all. If the attributes used are not correlated to the shift it seems unlikely that the selection would be good for OOD generalization.

- Unclear why other baselines like Öztürk et al. cannot be compared to, would the comparison be unfair?

Typos and other comments:
Maybe add a line defining the performance $P_j$ on page 3

line 428: outperform
line 905: we 9

**Questions:**

- How would you construct a meta-dataset in cases where the shift is not obvious or when the attribute is not labeled in the datasets?

- Does the coverage of the distribution over shifts impact results? For example, if the types of shifts are not equally represented in the meta-dataset or only some magnitudes of spurious correlation are represented.

- Would the space required to store the meta-dataset not get out of hand if you consider that a tunable model could have several entries with different values of one or several hyperparameters? Is there a way to mitigate this?

---

> ### Author Response · Authors · 2024-11-20
> **Response to Reviewer bNot**
>
> We thank the reviewer for the effort in reviewing our paper, and for the positive comments on the motivation and proposed solution.
>
> > **W1 & W2**: Choice of attributes. Need to know the attributes that are shifting a priori.
>
> Most prior work on OOD generalization does assume some knowledge on the attribute that is shifting, or of some labels of a spurious attribute. Some of our experiments therefore follow this setting.
>
> Taking a step beyond, **we also provide experiments on scenarios where the shifting attribute is not known**. Appendix E considers a practical setting where we assume no knowledge of the shifting attribute. We then perform a clustering in feature space to infer pseudo spurious attribute labels. Such ‘attribute pseudo-labels’ have been evaluated in prior work. Results in this setting indicate only a minor performance drop (Table 10).
>
> > **W3**. Why other baselines like Öztürk et al. cannot be compared to?
>
> We compared with Öztürk et al. in both controllable experiments (Table 1) and real-world dataset experiments (Table 6). This is the strongest baseline we can compare to, since there is no prior work in selecting learning algorithms for OOD generalization.
>
> > **Q1**. How to construct a meta-dataset when the shift is not obvious or when the attribute is not labeled in the datasets?
>
> There are two solutions provided in the paper, **both supported with experiments**.
>
> - The meta-dataset can be constructed from an existing dataset (such as CelebA) or a synthetic dataset where the spurious attribute is known. In Section 4, we experimented with an algorithm selector trained on a meta-dataset built from CelebA, then evaluated on target datasets built from COCO. Results indicate generalization across domains. I.e. the algorithm selector can be trained on a meta-dataset (with known attributes) and reused on another domain. This is made possible by the choice of dataset descriptors that are relevant across domains.
>
> - As mentioned above in this response, the spurious attribute can be inferred with a clustering method (Appendix E).
>
> We propose to better highlight these results in the paper.
>
> > **Q2**. Does the coverage of the distribution over shifts impact results?
>
> Yes, we believe the coverage of the distribution shifts would impact the results. In our experiments, we evaluate the algorithm selection on unseen datasets where either the types or magnitudes (or both) of distribution shifts are unseen when training the algorithm selector, and we observe that the algorithm selector generalizes well. The extent of this generalizability is an empirical question to be evaluated for different selectors' architectures, training objectives, etc.
>
> > **Q3**. Space required to store the meta-dataset?
>
> The reviewer is correct that the search space is larger when we consider algorithms plus a few hyperparameters, which consequently makes the meta-dataset larger. The most straightforward way of mitigating is to use less fine-grained values for hyperparameters, which sacrifices the performance but reduce the storage required. This trade-off between performance and computation/storage budget is unavoidable in most if not all systems, but in our setting the storage is utilized efficiently since each entry of the meta-dataset is only a fixed-length vector.

---

> > ### Comment · Reviewer_bNot · 2024-11-25
> > **Response to authors**
> >
> > I thank the authors for their rebuttal. Many of my concerns have been answered. However, the point about the attribute being known or not seems of larger importance than is being claimed. It need not be the case that the feature space can be clustered in such a way that it is simple to identify the pseudo-attributes. Overall, I would probably maintain my rating regarding this work, although I will await responses from the other reviewers.

---

> > > ### Author Response · Authors · 2024-11-25
> > >
> > > Thank you very much for the response, we are glad we addressed many of your concerns.
> > >
> > > Regarding the attributes:
> > > - We completely agree that the clustering heuristic is *not* universal to recover pseudo-attribute labels (none is, of course [1]). We include it as an evaluation of one popular heuristic.
> > > - We kindly highlight that *knowledge of the shifting attribute* is one common setup for OOD generalization. Many existing algorithms are based on this assumption. **We propose to clarify upfront in the manuscript that this is the primary setting we focus on.**
> > > - The study of the setting with known attributes **does not invalidate the whole paper**. Our contribution is to study whether OOD algorithm selection can be learned, given some characteristics of the data. The knowledge of the attribute is one choice we make for the initial setting in which we study this question, since it is a common setup as mentioned above.
> > >
> > > [1] ZIN: When and How to Learn Invariance Without Environment Partition? Lin et al. NeurIPS 2022.

---

> > > > ### Comment · Reviewer_bNot · 2024-11-26
> > > > **Response**
> > > >
> > > > I absolutely agree that the setting itself does not invalidate the paper as such, I made no such claim. My main point is that I feel that knowledge of the shifting attributes is a key point (and to some extent a limitation) and as such worth discussion, both here and in the paper. I hope to see some more engagement from other reviewers soon and will make a final determination after this. I thank the authors for their clear and concise responses.

---

> > > > > ### Author Response · Authors · 2024-11-26
> > > > >
> > > > > This is a great suggestion. We have updated the draft to explicitly reflect this point (see Section 1, Section 2, and the Discussion section; updates are marked in *orange*). Thanks again for your constructive feedback.

---

### Official Review · Reviewer_hewT · 2024-11-03

**Soundness:** 2
**Presentation:** 3
**Contribution:** 1
**Rating:** 3
**Confidence:** 3

**Summary:**

The authors propose an approach to algorithm selection for OOD generalization by treating it as a supervised classification problem over candidate algorithms. They introduce OOD-CHAMELEON, a system that constructs a dataset of datasets representing various distribution shifts and trains a model to predict the relative performance of algorithms based on dataset characteristics. The experiments demonstrate that OOD-CHAMELEON can outperform individual algorithms and simple selection heuristics on unseen datasets with controlled and realistic image data. The paper also inspects the model to reveal non-trivial data/algorithm interactions and conditions under which one algorithm might surpass another, offering insights into the applicability of existing algorithms.

**Strengths:**

Characterizing different OOD generalization tasks as distinct objectives and then selecting different methods to address those objectives is a reasonable approach. This strategy could have implications for the practical application of machine learning algorithms.

**Weaknesses:**

1. The contributions of this paper seem insufficient to me. The three proposed methods are all based on existing simple techniques and don't provide genuinely new insights into algorithm selection for different OOD problems.
2. While using a learning-based approach for method selection is a promising idea, this paper doesn't delve into a crucial aspect: why this problem is learnable in the first place. There's no discussion about the differentiability or continuity of the problem of selecting the optimal algorithm for different datasets. To me, it's a good direction, but the solution likely isn't straightforward. It might require some specialized design and deeper insights to properly address the inherent challenges of learning algorithm selection.
3. Although the paper is complete in content, the layout of some parts is slightly compact, especially the algorithm description and experimental parts, which lack a clear modular structure. Some paragraphs are too long and the reading experience is poor. If the algorithm description, experimental design, and result analysis are divided into clearer sub-modules, the reading fluency may be improved.

**Questions:**

1. How efficient is OOD-CHAMELEON in processing large-scale data sets?
2. Are there any further optimization solutions to improve its efficiency in practical applications?

---

> ### Author Response · Authors · 2024-11-20
> **Response to Reviewer hewT**
>
> We thank the reviewer for the effort in reviewing our paper, and for acknowledging the significance and practical value of this new problem.
>
> > **W1**. The three proposed methods are all based on existing simple techniques. No new insights into algorithm selection for different OOD problems.
>
> There were **no real insights about OOD algorithm selection in the existing literature**. Mutliple works (L70-71) called the need for work on this topic as a future direction, but no advance had been made.
>
>
> We claim as novel are:
> (1) evaluating the feasibility of algorithm selection for OOD generalization;
> (2) designing a proof-of-concept method (OOD-Chameleon).
>
> As examples of novel insights, we show that:
> - the algorithm selection is learnable in a data-driven way from a collection of datasets;
> - the effectiveness of algorithm selection depends on the parametrization of the selector;
> - there exist non-trivial relationships between data and algorithms that can't be captured e.g. by a linear model;
> - the training of an algorithm selector with a classification objective is much more effective than a regression, even though the latter is conceptually more straightforward;
> - examining the trained selector indicates which characteristics of a dataset make some algorithms preferred over others (Section 3.4).
>
> None of these findings were obvious or simple, neither are the design and results of (2) above. **We would like to know from the reviewer which exact contributions aren't novel enough or too simple, with precise references to prior work**.
>
> > **W2**. Why is the problem learnable?
>
> The algorithm selection is made learnable by turning it into a supervised problem, and learning from a *collection of datasets*. This **meta learning** approach turns the algorithm selection into a regime where classical results from statistical learning theory applies.
> We propose to make this clearer. Please also see the discussion in Section 2.1 ("Is the Selection of the Best Learning Algorithm Even Possible?") including the relevance of the no-free lunch theorem.
>
> We also refer to prior works [1-4] (Section 5) on the analogous problem of **learning-based** selection of pretrained models (rather than OOD algorithms) that is also approache in an **empirical and data-driven** manner.
>
> [1] Quick-Tune: Quickly Learning Which Pretrained Model to Finetune and How. Arango et al. ICLR 2024
> [2] Zero-shot AutoML with Pretrained Models. Öztürk et al. ICML 2022
> [3] TASK2VEC: Task Embedding for Meta-Learning Achille et al. ICCV 2019
> [4] Model Spider: Learning to Rank Pre-Trained Models Efficiently Zhang et al. NeurIPS 2023
>
>
> > **Q1 & Q2**. How efficient is OOD-Chameleon?
>
> The method is both time- and space-efficient.
> - **Time**: the trained algorithm selector only takes the characteristics of the target dataset as input and directly predicts the suitable algorithm, and the algorithm selector is parametrized by MLPs.
> - **Space**: each entry of the meat-dataset only consists of the dataset descriptor of size l, the one-hot algorithm vector of size M and the corresponding scalar OOD performance, which means the storage only grows linearly (this is most likely very light because l and M is a small to moderate constant).

---

### Official Review · Reviewer_uSdg · 2024-11-03

**Soundness:** 1
**Presentation:** 1
**Contribution:** 2
**Rating:** 3
**Confidence:** 4

**Summary:**

Authors formulate a new problem setting of predicting the performances of multiple algorithms on a given dataset without training models. They propose a framework called OOD-CHAMELEON to address this setting.

**Strengths:**

The problem setting is novel.

**Weaknesses:**

1. The practice of predicting performance with only some simple characteristics of the dataset and an algorithm represented by a one-hot vector does not really make sense to me. Given a learning algorithm and a dataset, there are too many other factors that could largely influence the final trained model, including hyperparameters. However, all other factors are ignored in this setting.
2. Experiments are not sufficient enough to support the claims made in the paper.
   - Only five algorithms are selected as candidates while there are various OOD algorithms besides them. I believe at least several SOTA or influential algorithms could be added.
   - Only two real-world datasets are included.

3. The writing seems poor.
   - The writing seems awkward in some places.
     - In Line 018 "treats the task as a supervised classification"
     - In Line 025 "on unseen datasets of controllable and realistic image data".
     - In Line 110, "It consists in predicting the best strategy to obtain a robust model given a statistical descriptor of the dataset." Here "consist in".
   - The writing seems imprecise/ambiguous in some places.
     - In Line 052 "A well-known study by Gulrajani & Lopez-Paz (2020) showed that none of these methods surpasses an ERM baseline across a collection of datasets." If here "these methods" refer to the upper line, then "the more complex ones" in Line 051 are not included since there are many new algorithms proposed in recent three years that are not included in the paper of DomainBed.
     - In Line 075 "We posit that OOD generalization can be improved if we knew which algorithm to apply in each situation" why use "knew" instead of "know" here?
     - In Line 105, "our findings call for improving OOD generalization by learning to better apply existing algorithms, instead of designing new ones." Here "instead of designing new ones" might be interpreted as designing new algorithms is less useful than learning to apply existing algorithms.
   - The notations are not clear enough and some are abused.
     - There are multiple confusions of variables and sets. For example, in Section 2.1, $X$ seems to be a random variable, however in $x\in X$, it seems to be the support set of the input variable. In Section 2.2, $\mathcal{A}(\cdot):D^{tr}\rightarrow h_{\theta}$, when defining a function mapping, it should be between two sets. However, here $D^{tr}$ is a dataset instead of the data space and $h_{\theta}$ is a detailed hypothesis instead of a hypothesis space.
     - In Line 130, is $X_c$ a subset of variables of the input? This should be clarified.
     - In Line 206, $|\mathcal{G}\_i|=n_{te}/i$. I suppose here $i$ is a wrong notation, which should be replaced by the number of groups.

**Questions:**

1. In this paper, distribution shifts are categorized into three categories. However, any distribution shift can be decomposed into covariate shift and concept shift. Why is label shift required when shift on $P(X)$ and $P(Y|X)$ both exist?
2. In Line 132, "A shift of spurious correlations implies a variation of an attribute/label co-occurrences, which means a shift on $P(Y|A)$ but not $P(Y|X_c)$." Previously the spurious correlation is stated as shift of $P(Y|X)$. However, if $X_c$ is a subset of variables of $X$, and if there is no shift on $P(Y|X_c)$, there should not be shift on $P(Y|X)$ since $P(Y|X)=P(Y|X_c)$.
3. In Equation 2, why is the positive correlation between $y$ and $a$ considered as spurious correlation, while the negative correlation is not considered? In other words, when $d_{sc}$ is close to 1 or close to 0, both circumstances could imply strong spurious correlations.

---

> ### Author Response · Authors · 2024-11-20
> **Response to Reviewer uSdg**
>
> We thank the reviewer for the effort in reviewing our paper, and for acknowledging the novelty of the problem (algorithm selection for OOD generalization).
>
> > **W1.1**. Using simple characteristics of the dataset and an algorithm represented by a one-hot vector does not really make sense.
>
> **These design choices are supported by prior work**. See [1,2] about the analogous problem of selecting a pre-trained model to fine-tune on a target dataset. These works use even simpler characteristics of the dataset as input (e.g. number of classes and channels).
>
> The choice of **one-hot encodings** to represent algorithms is an irrelevant implementation choice. The criticism misses the fact that the purpose of our approach is to *learn* about each algorithm from examples. For example, [1,2] also use one-hot encodings to represent pre-trained models. This choice may have been suboptimal because of possible known relations (e.g. similar architecture) that could have been encoded otherwise. This is not the case in our setting. **We would like to know how the one-hot representations are problematic in the reviewer's eye** to clarify this in the paper.
>
> [1] Quick-Tune: Quickly Learning Which Pretrained Model to Finetune and How. Arango et al. ICLR 2024
> [2] Zero-shot AutoML with Pretrained Models. Öztürk et al. ICML 2022
>
> > **W1.2**. Hyperparameters are ignored in this setting.
>
> A straightforward extension is to include necessary hyperparameters (such as learning rate) to the search space (mentioned at L511), and then the algorithm selector is trained to predict the optimal algorithm along with its suitable hyperparameters.
>
> > **W2**. Experiments not sufficient to support the claims
>
> We were very careful in making claims supported by concrete evidence. Hence **we would appreciate a precise statement of the problematic claims** from the reviewer, so as to tone them down or clarify the evidence in a revision.
>
> The fact that more datasets/models/algorithms can be evaluated is always true. Remember that this paper introduces a **new setting and a whole new take on the field of OOD generalization** which has been stagnant for a couple of years.
>
> We propose to clarify upfront (in the abstract) that the paper does not provide a new off-the-shelf solution. It opens a new research direction, and the main claim is about evaluating whether OOD algorithm selection is viable as a learning problem (cf. the title). Our experiments are designed to support this claim by using proven methods known to address specific types of distribution shifts, which are actually used in deployments of ML (not only in one-off academic papers).
>
> The experiments are also designed to support our claims by using controlled conditions. The setups based on COCO and CelebA were deemed most suitable, but we welcome suggestions that we could add to the final version of the paper. Extensions to other algorithms/datasets are clearly stated as new avenues opened up by this paper.
>
>
> > **Q1**. Why is label shift required when shift on P(X) and P(Y|X) both exist?
>
> The reviewer is correct that a shift in $P(Y)$ manifests indirectly in $P(X)$ or $P(Y|X)$. However modeling $P(Y)$ directly is preferred in practice for efficiency and clarity in the analysis and generation of data, when there is a need to focus on label distribution. See e.g. [3,4].
>
> [3] Change is Hard: A Closer Look at Subpopulation Shift. Yang et al. ICML 2023
> [4] A Unified View of Label Shift Estimation. Garg et al. NeurIPS 2020
>
> > **Q2**. Previously the spurious correlation is stated as shift of P(Y|X). However, if Xc is a subset of variables of X, and if there is no shift on P(Y|Xc), there should not be shift on P(Y|X) since P(Y|X)=P(Y|Xc).
> >
>
> The equality $P(Y|X)=P(Y|X_c)$ holds only if
> - $X_c$ is sufficient for predicting Y (which is satisfied), and
> - features in $X$ \ $X_c$ are conditionally independent of $Y$ given $X$ (which is **not** satisfied because $A$ is spuriously correlated with $Y$).
>
> We will make this clearer in the paper.
>
> > **Q3**. In Equation 2, why is the positive correlation between y and a considered as spurious correlation, while the negative correlation is not considered?
>
> We did consider it and defined the range appropriately in [0,1] (L203, below Equation 2).

---

### Author Response · Authors · 2024-11-20
**General response**

We are grateful for the reviewer's time and are taking the suggestions into account to improve the manuscript. All reviewers acknowledged the importance of the new research direction of algorithm selection for OOD generalization. This work provides a **new perspective** on OOD generalization by exploring the potential of existing algorithms.

We have provided detailed responses to each reviewer and summarize a few shared concerns below:

- (Reviewers 5F9j & hewT) *Theoretical analysis required.*
    - The whole point of this paper is the possibility of going beyond existing theoretical chracterizations of shifts.  Many theoretical results have limited real-world applicability because the data often contains **mixtures of different types of shifts**. Our meta-learning approach turns algorithm selection into a supervised problem where classical results from statistical learning theory applies.
    - There are many prior works (see Section 5) that tackle pre-trained model selection in a similar **empirical and data-driven** manner.
- (Reviewers 5F9j & uSdg) *More algorithms/datasets/models required.*
    - The fact that more evaluation can be done is always true.
        - Algorithms: the chose algorithms are *proven* methods to address different distributions shifts, which are actually **used in deployments** of ML (not only in one-off academic papers).
        - Datasets: (1) We first use synthetic experiments to validate the feasibility of the setup with full control over the data-generating process. (2) CelebA and COCO were deemed most suitable for a realistic yet controlled evaluation of the method.

---

> ### Author Response · Authors · 2024-11-25
> **Updated manuscript**
>
> We thank again the reviewers for their efforts. We used the feedback to improve the manuscript with additional experiments, clarifications and details, and we enhanced the overall readability. Major updates are in blue in the PDF:
> - Additional training details;
> - Clarification of the notations;
> - Clarifications of the motivations;
> - Rationale for the choice of algorithms;
> - Clarification of limitations and avenues for future work;
> - New experiments (Appendix F, Table 11) on more dataset (Colored-MNIST, for its sufficiency of samples in each group) in response to the ‘limited datasets’ weakness. The algorithm selector is trained on the meta-dataset constructed from CelebA and evaluated on 150 unseen datasets. These experiments further validate that the algorithm selector can be **trained once** on a meta-dataset of synthetic distribution shifts, and then **reused on new unseen datasets**.
>
> ---
>
> **We will be grateful to the reviewers for sharing whether our efforts have addressed their concerns, and improved their assessment of this work.**

---

### Meta-Review · Area_Chair_9DHF · 2024-12-13

**Metareview:**

The overall quality is not good enough to make it an ICLR paper.

The authors should not complain all the three negative reviewers (rating 3) and say the only positive reviewer (rating 6) is the only one providing reasonable comments and suggestions, which is too aggressive and does not help anything in the end.

If I were a reviewer, I would think formalizing algorithm selection as supervised classification over candidate algorithms itself is problematic, since those algorithms are not designed to compete with each other just to win the selection, while single-label multi-class problems always have classes strongly competing with each other.

**Additional Comments On Reviewer Discussion:**

The rebuttal didn't address the concerns from the reviewers.

---

### Decision · Program_Chairs · 2025-01-22

Reject